# i-Motif folding intermediates with zero-nucleotide loops are trapped by 2'-fluoroarabinocytidine via F⋯H and O⋯H hydrogen bonds

Roberto El-Khoury [1], Veronica Macaluso[2], Christopher Hennecker[1], Anthony K. Mittermaier[1], Modesto Orozco[2], Carlos González [3], Miguel Garavís [3,4]✉ & Masad J. Damha [1,4]✉

G-quadruplex and i-motif nucleic acid structures are believed to fold through kinetic partitioning mechanisms. Such mechanisms explain the structural heterogeneity of G-quadruplex metastable intermediates which have been extensively reported. On the other hand, i-motif folding is regarded as predictable, and research on alternative i-motif folds is limited. While $TC_5$ normally folds into a stable tetrameric i-motif in solution, we report that 2'-deoxy-2'-fluoroarabinocytidine (araF-C) substitutions can prompt $TC_5$ to form an off-pathway and kinetically-trapped dimeric i-motif, thereby expanding the scope of i-motif folding landscapes. This i-motif is formed by two strands, associated head-to-head, and featuring zero-nucleotide loops which have not been previously observed. Through spectroscopic and computational analyses, we also establish that the dimeric i-motif is stabilized by fluorine and non-fluorine hydrogen bonds, thereby explaining the superlative stability of araF-C modified i-motifs. Comparative experimental findings suggest that the strength of these interactions depends on the flexible sugar pucker adopted by the araF-C residue. Overall, the findings reported here provide a new role for i-motifs in nanotechnology and also pose the question of whether unprecedented i-motif folds may exist in vivo.

[1] Department of Chemistry, McGill University, Montréal H3A0B8, Canada. [2] Institute for Research in Biomedicine (IRB Barcelona), Barcelona 08028, Spain. [3] Instituto de Química Física 'Rocasolano', (IQFR-CSIC), Madrid 28006, Spain. [4] These authors jointly supervised this work: Miguel Garavís, Masad J. Damha. ✉email: mgaravis@iqfr.csic.es; masad.damha@mcgill.ca

**K**inetic partitioning mechanisms are believed to govern the folding of quadruplex nucleic acid structures. These mechanisms support the formation of multiple, coexisting conformations with the same sequence, differing in their thermodynamic stability. G-quadruplexes, in particular, are notorious for their structural polymorphism resulting from folding landscapes with several energy minima[1]. On the other hand, i-motifs, which consist of two parallel duplexes intercalated by cytosine:cytosine (C:C$^+$) base pairs, also seem to fold through kinetic partitioning mechanisms[2–8] but are deemed less structurally heterogenous[9–11]. Apart from intermediates of unknown structure or i-motif conformers differing in their C:C$^+$ stacking order[3–7,12], no other alternative i-motif folding intermediates have been isolated and characterized to date.

Here, we report the first high-resolution structure of a metastable dimeric i-motif intermediate, thereby expanding the scope of i-motif folding landscapes. This finding is timely owing to the relevance of i-motifs to disease (regulating DNA transcription, protooncogene transcription, and telomere homeostasis)[13–19], as well as their nanotechnological applications (molecular rotors[20], cellular sensors[21] and hydrogel components[22]) which rely on their pH-dependent folding.

Oligonucleotide d(TC$_5$) forms a well-characterized tetrameric i-motif in solution, which represents the first i-motif structure ever reported[23]. Studies have suggested that the tetrameric structure assembles through a sequential folding pathway with dimeric and trimeric intermediates that have evaded isolation[2]. Isolating the off-pathway dimeric i-motif intermediate reported here is only possible once four (Fig. 1, ON4a-e) or all five (Fig. 1, ON5) 2′-deoxycytidine (dC) residues are substituted with 2′-deoxy-2′-fluoroarabinocytidine (araF-C) (Fig. 1). We have previously shown that substituting dC with araF-C in i-motif-forming sequences dramatically slows down i-motif unfolding kinetics[24]. In line with these previous findings, we show here that araF-C can also stabilize non-native structures, allowing the observation of i-motif folding intermediates. We had also demonstrated that araF-C substitutions superlatively increase the thermal stability of i-motifs at both acidic and neutral pH[24–26]. However, elucidating the structural reasons for this stabilization has been precluded because of the absence of model systems with sufficiently high NMR resolution required for quantum mechanics-based computational methods. Therefore, in addition to characterizing its folding, we determine that intra-residual

organic fluorine- and oxygen-hydrogen bonds contribute to the stability of the structure, which persists in solution for months.

## Results

**Isolating the dimeric ON5 species.** Sample preparation influences the nature of the araF-C-modified ON5 species formed in solution. Non-denaturing gel electrophoresis reveals that, at higher concentrations (500 μM), slow annealing (SA) (see Methods) leads to the formation of a low-mobility species, previously characterized as the tetrameric T(araF-C)$_5$ i-motif[26], and a small population of a higher-mobility species that we show here to be an off-pathway dimeric i-motif (Fig. 2A). Interestingly, rapid (snap-cool) annealing (RA) results in a band with intermediate mobility, presumably corresponding to a trimeric species or a less-structured dimer. At concentrations below 200 μM, ON5 still forms a mixture of species under SA but forms the high-mobility dimeric species exclusively under RA (Fig. 2A). In contrast, unmodified TC$_5$ (ON0) fails to form the high-mobility species regardless of the annealing method (Supplementary Figure 1). As expected, it forms the low-mobility tetrameric i-motif under SA and, surprisingly, forms an additional species of intermediate mobility under RA, similar to that which forms in ON5 at high concentrations and RA (Supplementary Figure 1).

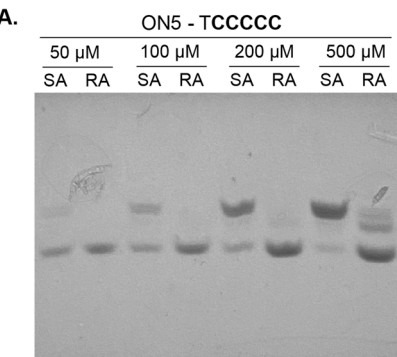

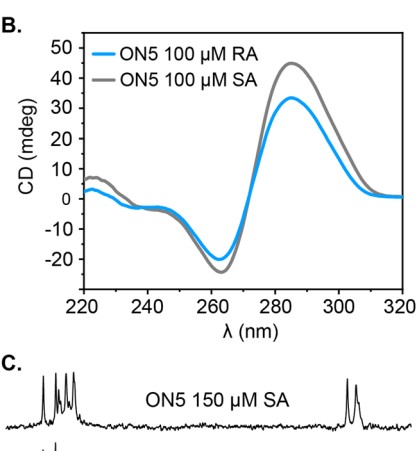

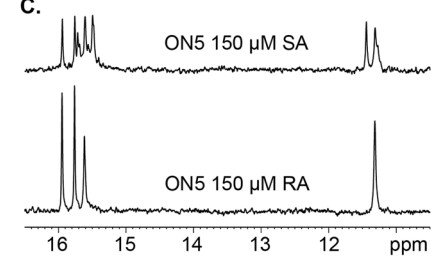

**Fig. 2 Influence of oligonucleotide concentration and annealing method on ON5 structural populations. A** Non-denaturing gel electrophoresis of ON5 across different concentrations upon RA and SA. **B** CD spectra of 100 μM RA (blue) and SA (gray) ON5 at 5 °C. **C** $^1$H NMR imino signals of 150 μM RA ON5 and SA ON5 at 5 °C. Buffer conditions are aqueous 10 mM NaP$_i$ pH 5 (10% D$_2$O for NMR spectra).

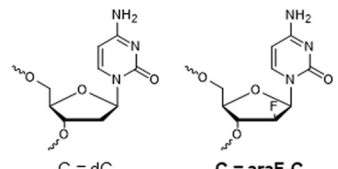

C = dC          **C** = araF-C

| Oligonucleotide Code | Sequence (5′-3′) |
|---|---|
| ON0 | TCCCCC |
| ON5 | TCCCCC |
| ON4a | TCCCCC |
| ON4b | TCCCCC |
| ON4c | TCCCCC |
| ON4d | TCCCCC |
| ON4e | TCCCCC |
| HC3 | TCCTTTTCCA |

**Fig. 1 Oligonucleotide sequences used in this study.** i-Motif forming oligonucleotides are listed in the table, and the chemical structures of dC and the substituent araF-C are represented above the table.

**Determining the structure of the dimeric ON5 species**. We used circular dichroism (CD) and NMR spectroscopy to characterize the structure of the isolated dimeric ON5 species. Regardless of annealing conditions, the CD spectra of ON5 (Fig. 2B) display the same maximum and minimum at around 285 nm and 265 nm, respectively, and only differ in signal intensity. These spectral features are characteristic of i-motifs and are similar to those of the tetrameric ON0 i-motif (Supplementary Figure 2)[26]. Moreover, both RA and SA ON5 exhibit signals in the 15-16 ppm region of the [1]H NMR spectrum (Fig. 2C), which are characteristic of imino protons involved in C:C$^+$ base pairs. While the SA sample shows multiple, overlapping signals, the RA sample shows three sharp and well-dispersed signals, consistent with the formation of a single species and in agreement with observations from non-denaturing electrophoretic experiments. Similarly, RA ON5 shows a single sharp signal at 11.3 ppm, which is characteristic of a T:T base pair, while SA ON5 exhibits several. In assessing the stability of the dimeric RA ON5 species, we found that most imino signals remain visible at pH 7.0 (Supplementary Figure 3), thereby demonstrating the stabilization incurred by araF-C residues.

To determine the structure of RA ON5, 2D NMR spectroscopy was used. The NOESY and TOCSY spectra (Fig. 3A) confirmed the presence of a single species, particularly through our observations of a single cross-peak between the methyl protons of the thymine of residue T1 and its aromatic proton, as well as five intense cross-peaks between the aromatic protons (H5 and H6) of each of the araF-C (**C2** through **C6**). While only three imino-amino cross-peaks of C:C$^+$ homo-base pairs are observed between 15 and 16 ppm (Fig. 3A), the amino protons of all five araF-C residues appear in the range of chemical shifts characteristic of protonated cytosines (8.2–9.5 ppm) (Supplementary Table 1), suggesting they are all involved in base-pairing. The absence of other sequential cross-peaks and a large number of sugar-sugar NOEs ruled out the possibility of parallel duplex formation. To assign **C3** through **C6**, we recorded the 1D [19]F-NMR spectra (Fig. 3B) of a series of oligonucleotides, ON4a-e (Fig. 1), which is based on the sequential substitution of one araF-C of ON5 with a dC residue. Non-denaturing gel electrophoresis (Supplementary Figure 4), CD (Supplementary Figure 5) and [1]H-NMR (Supplementary Figure 6) spectra of all the oligonucleotides indicate that they adopt the same conformation as ON5 when subjected to RA. The 1D [19]F-NMR spectrum of ON5 shows five sharp and dispersed signals in a range of –188 to –202 ppm (Fig. 3B) corresponding to each of the five [19]F nuclei in the molecule. As expected, the spectra of ON4a-e show four signals, meaning that each missing fluorine signal could be assigned to the araF-C residue of ON5 that has been substituted with deoxycytidine. In ON4b specifically, two [19]F signals appear close together at −193.10 and −193.14 ppm and are resolved in the [1]H-[19]F-HOESY (Supplementary Figure 7). After assigning each of its fluorine signals (Supplementary Table 1), the [1]H-[19]F-HOESY spectrum of ON5 (Fig. 3C) was used to identify the H2′ protons through their intense intra-residual cross-peaks with [19]F, thereby facilitating the complete, unambiguous assignment of **C3**-**C6** residues. Consequently, we identified key i-motif minor groove NOE cross-peaks (H1′-H1′ between stacked cytosines and reciprocal H2′-H1′ between stacked cytosines facing their 3′ edges), which serve as conclusive evidence that RA ON5 folds into a dimeric i-motif with the stacking order **C2-C6-C3-C5-C4** (top to bottom) (Fig. 4A).

**Structural features of the dimeric ON5 i-motif**. We used 50 structurally-relevant distance restraints (Supplementary Table 2) to resolve the structure of the dimeric i-motif. These are derived

from NOESY and [1]H-[19]F-HOESY spectra (see Methods) and torsion angular restraints obtained from qualitative analysis of DQF-COSY (Supplementary Figure 8).

H2′-H1′ cross-peak intensities in the COSY spectrum indicate that araF-C sugars of **C3** and **C5** adopt a north conformation, T1, **C2** and **C4** a south conformation, and **C6** a Northeast conformation. Supplementary Table 3 displays the average pseudorotation parameters of the dimeric ON5 i-motif and Supplementary Table 4 displays its average dihedral angles and order parameters.

The dimeric i-motif is formed by two tightly turned strands, associated head-to-head, and stabilized by five C:C$^+$ and one T:T inter-strand homo-base pairs. The change in strand orientation occurs through an unprecedented nucleotide-free loop at the **C4**-p-**C5** step (Fig. 4A). Interestingly, the T:T base pair does not follow the alternating intercalation pattern of the C:C$^+$ base pairs, so the intercalated unit of the structure consists of five C:C$^+$ base pairs, with **C2**:**C2**$^+$ and **C4**:**C4**$^+$ being the most external ones. Importantly, it was possible to detect HOE cross-peaks correlating the fluorine atom of **C3** with the amino protons of **C5** and vice versa (Fig. 3C). The same correlations are observed between **C2** 2′F and **C6** amino protons, but they are not observed between **C6** 2′F and **C2** amino protons, likely because the **C2**:**C2**$^+$ base pair is the most external, with its amino protons exchanging more rapidly with solvent. We also detect intense HOE cross-peaks correlating 2′F and H6 aromatic protons of all araF-C residues (Fig. 3C). Overall, the presence of these HOE correlations is indicative of short distances between 2′F atoms and inter-strand amino protons or intra-residual aromatic protons (Fig. 4B). These results led us to investigate the formation of 2′F···H hydrogen bonds, which we have previously postulated to be important in araF-C stabilized i-motifs[24–26].

**F···H and O···H bonds contribute to the stability of araF-C-modified i-motifs**. The establishment of organic fluorine-hydrogen bonds in nucleic acid systems has seldom been conclusive, so we resorted to experimental (NMR) and computational methods to determine their existence in the dimeric i-motif. We found that all araF-C aromatic proton (H6) signals of RA ON5 exhibit significant broadening in linewidth in the [19]F-coupled spectrum (Fig. 5A). Given the long through-bond distance (5 bonds) between the 2′F and H6 atoms (Fig. 4B), such dramatic broadening is unambiguously attributed to a through-space scalar coupling occurring between these atoms. **C3**H6, in particular, exhibits clear splitting in the [19]F-coupled [1]H NMR spectrum and resonates as a doublet of doublets, allowing us to estimate the magnitude of the J coupling of the H-bond (≈3.8 Hz) (Fig. 5A).

To determine whether other araF-C stabilized i-motifs also feature intra-residual fluorine-hydrogen bonds, we performed [19]F-coupled and decoupled [1]H-NMR on SA ON4b (Fig. 1; Supplementary Figure 9A), which folds exclusively into a tetrameric i-motif at high concentrations (Supplementary Figures 4 and 6). Due to the larger size of the tetramer, the linewidth of the signals is markedly broader than that of the dimer. Nevertheless, specific differences in the fine structure of the aromatic signals of the araF-C residues are detected between the two spectra. Again, all—and only—the aromatic proton signals of araF-C broaden when [19]F-coupling is enabled. Similar results (Supplementary Figure 9B) are obtained for the previously-studied HC3 centromeric sequence (Fig. 1), which forms a dimeric i-motif[26,27]. Consequently, we infer that intra-residual 2′F···H6 hydrogen bonding contributes to the enhanced stability of i-motifs containing araF-C.

We next sought to understand whether the sugar pucker adopted by the araF-C residues could influence the magnitude of

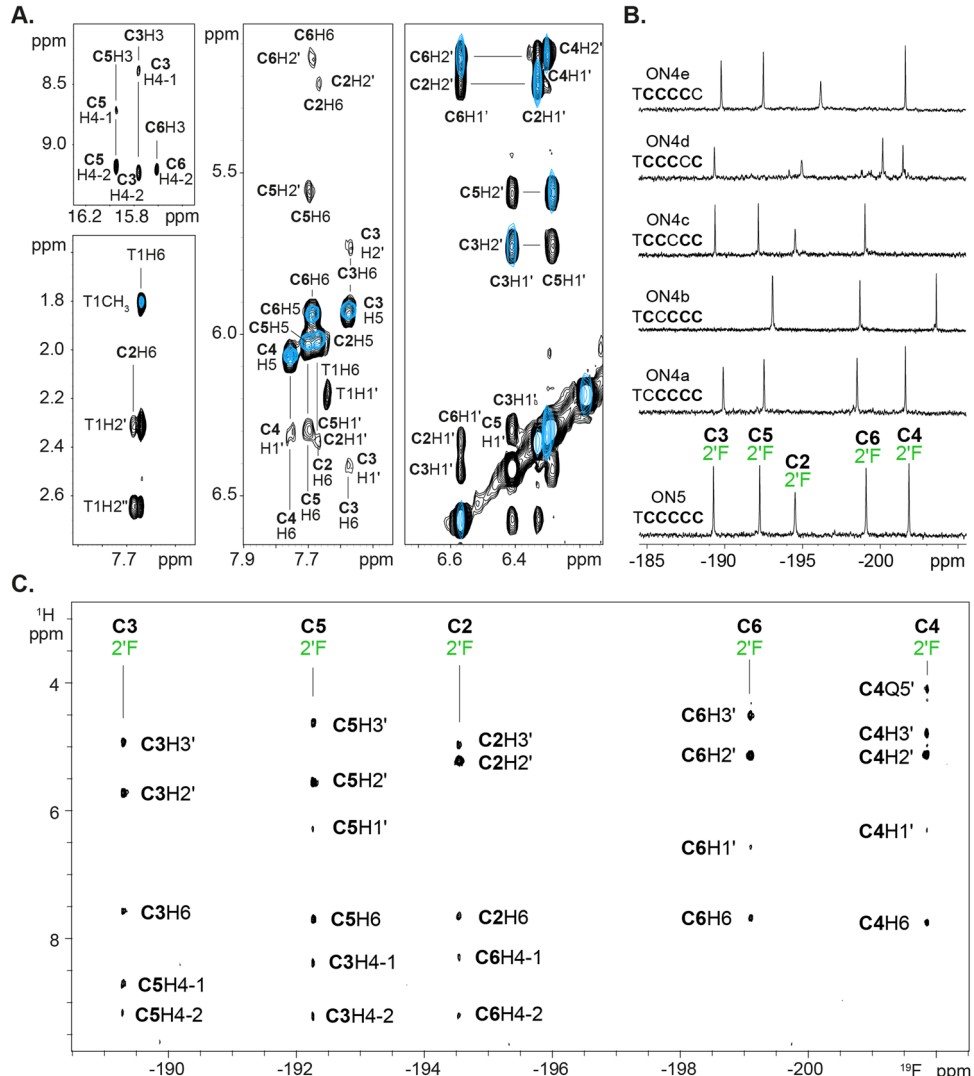

**Fig. 3 Characterization of ON5 dimeric structure by NMR. A** Overlapping NOESY (black) and TOCSY (blue) spectra corresponding to RA ON5. Top left: imino-amino cross-peaks. Bottom-left: cross-peaks between methyl protons of T1 and its own aromatic proton. Sequential cross-peaks between H2′/H2′′ of T1 and H6 of **C2**. Middle: five intense TOCSY signals correlating H5 and H6 of the five araF-C residues. Low-intensity intra-residual H1′-H6 and H2′-H6 cross-peaks. Right: Inter-residual H1′-H1′ and H1′-H2′ cross-peaks across the minor groove. **B** 1D $^{19}$F NMR spectra of RA ON5 and RA ON4a-ON4e. **C** $^{1}$H-$^{19}$F HOESY spectrum of RA ON5. All spectra were acquired with 150 µM RA ON5 in 10 mM NaP$_i$ pH 5, at 5 °C.

the through-space $J$ constant and therefore the strength of the H-bond. Just like **C3**, **C5** adopts a north (C3′-endo) sugar pucker (Fig. 4C). However, the H6 signal of **C5** overlaps with other signals in the $^{19}$F-coupled spectrum (Fig. 5A), making the evaluation of the $J$ constant difficult. This obstacle is overcome in the 1D $^{1}$H-NMR spectrum of the ON4b oligonucleotide (Fig. 5B), which also folds into the same dimeric i-motif upon RA (Supplementary Figure 7). In the spectrum, the H6 signal of **C5** appears isolated and splits in the $^{19}$F-coupled spectrum with the same J coupling of 3.8 Hz as the signal of **C3**H6 in the RA ON5 sample. Given that **C3** and **C5** are the only residues that adopt a north (C3′-endo) sugar pucker (Supplementary Figure 8 and Supplementary Table 3), we postulate that i-motif araF-C residues with a north conformation exhibit intra-residual H-bonds of higher strength than those with other conformations.

To complement our experimental findings on 2′F···H bonding, we used the NCI (non-covalent interaction) computational method, which is effective in detecting weak interactions, such as intra-residual hydrogen bonds[28] or those involving a fluorine acceptor[29]. The NCI framework is based on the analysis of the

reduced density gradient (s) across a molecule, which is plotted as function of sign($\lambda_2$)$\rho$, where $\rho$ is the electronic density and $\lambda_2$ the second eigenvalue of $\rho$ Hessian. In regions of hydrogen bonds, sign($\lambda_2$)$\rho$ would have low negative values while s would be close to 0[30,31]. Accordingly, we constructed NCI plots (Supplementary Figures 10–13) for a total of 20 structures consisting of models of two capped cytosines (**C2**-**C6′**, **C3**-**C5′**, **C6**-**C2′**, and **C5**-**C3′**) (Supplementary Figure 14) extracted from NMR-restrained molecular dynamics simulations of the dimeric i-motif. To associate molecular interactions with spikes in the NCI plots, we chose a representative structure for each of the two-residue models and reported the relative NCI plot (Fig. 5C) along with the 0.04 s isosurface in the molecular representation (Fig. 5D). Additionally, NCI plots of single-residue models (Supplementary Figures 15–18) were used to discern intra- from inter-residual interactions (a detailed assignment is discussed in the Supplementary Information).

Using the NCI method, we detected intra-residual fluorine-hydrogen bonds involving **C2**, **C3**, **C5**, and **C6** residues and their protonated (′) equivalents (Fig. 5D), thereby corroborating our

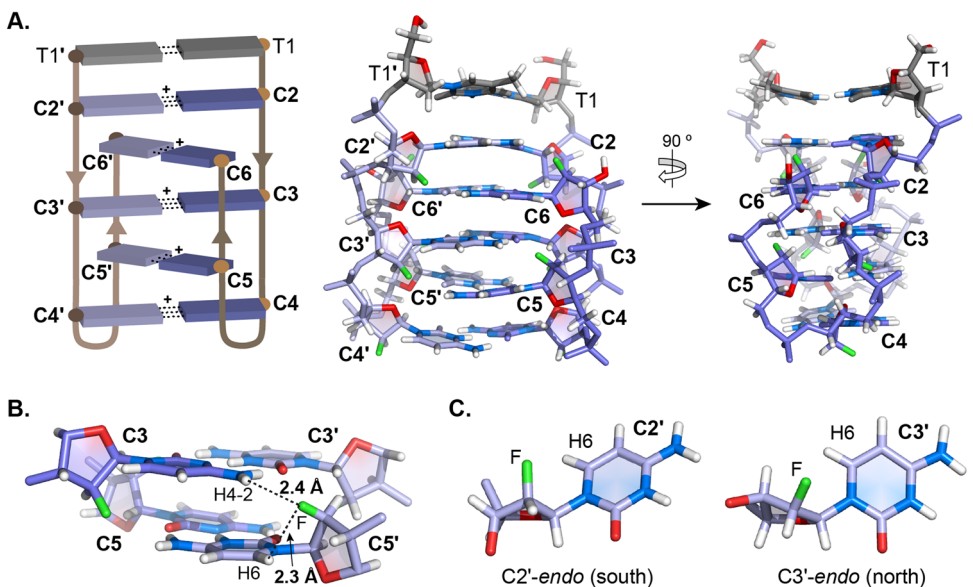

**Fig. 4 ON5 dimeric i-motif structural characteristics. A** Schematic view (top left), and three-dimensional structure viewed from the major groove (top middle) and minor groove (top, right). **B** 3′-3′ interphase across the major groove showing inter- F···H-amino and intra-residual F···H6 contacts. **C** Structures of **C2′** and **C3′** residues with C2′-*endo* and C3′-*endo* sugar pucker, respectively. dT and araF-C are in gray and slate blue, respectively. Atom color code: fluorine: green; nitrogen: blue; oxygen: red; hydrogen: white. Atom type is not color-differentiated along the backbone. ′ symbols denote residues of the strand with protonated araF-Cs.

NMR-based results showing the splitting of araF-C H6 aromatic signals upon $^{19}$F coupling (Fig. 5A, B). In each of the pairs of residues examined, we also detected inter-residual fluorine-hydrogen bonds between 2′F and cytosine H4 amino protons, as well as intra-residual O2···H1′ and O4′···H6 hydrogen bonds. The networks of hydrogen bonds detected likely contribute to the stabilization of i-motifs upon modification with araF-C.

**The dimeric ON5 i-motif is an off-pathway kinetic trap**. We conducted thermal hysteresis (TH) experiments using a temperature-controlled UV spectrometer to understand the folding of the dimeric T(araF-C)$_5$ i-motif and compare it to the folding of the well-characterized tetrameric i-motif. In TH experiments, the temperature ramp rate is fast compared to the folding and unfolding rates of the sample, leading to melting and annealing traces that shifted to higher and lower temperatures, respectively. Two scan rates were applied to ON0 and ON5 samples at two concentrations (50 and 250 µM), representing either RA (5 °C/min) or SA (0.5 °C/min). The resulting TH profiles (Fig. 6A) show biphasic melting curves and pronounced hysteresis, suggesting that both oligonucleotides can form dimeric and tetrameric species with lower- and higher-temperature melting transitions, respectively[26].

The tetrameric ON0 i-motif was previously proposed to fold through sequential monomer additions, proceeding through dimer and trimer intermediates[2]. Owing to its folded nature, we hypothesized that the dimeric ON5 i-motif we isolated was an off-pathway, kinetically-trapped intermediate resulting from a structural rearrangement of the on-pathway dimeric intermediate. To test this hypothesis, the TH traces for ON0 and ON5 were globally fit to two different assembly pathways (Fig. 6B): (1) following the sequential intermolecular assembly mechanism and (2) allowing for an off-pathway dimeric species. The dimer spectral absorption coefficient was assumed to be half-way between those of the monomer and tetramer, as raw absorbance changes were found to be smaller in magnitude when the dimer was formed. The improvements in RSS given by mechanism 2, which accounts for the formation of an off-pathway dimer,

compared to mechanism 1, was calculated using F-test statistics[32] and found to be significant at levels of $p \leq 10^{-2}$ for ON5, whereas ON0 did not show any improvement to the RSS for mechanism 2 (Supplementary Table 5).

Furthermore, by modeling the assembly of the tetrameric and dimeric i-motifs from TH experiments, we can extract information on the populations of individual species in the sample. For instance, isothermal kinetic simulations of RA ON0 and ON5 at low concentrations (50 µM) show a large degree of dimer assembly at 4 °C (Supplementary Figure 19), which is consistent with previous experiments showing that the rate-limiting step of tetramer assembly for ON0 is the formation of the trimeric species[2]. In the case of ON0, this dimer intermediate quickly converts to the tetrameric species. In contrast, the ON5 on-pathway dimeric species folds into the off-pathway dimeric i-motif which acts as a kinetic trap and further slows the assembly of the tetrameric species and remains folded for days at low temperatures.

## Discussion

We have previously shown that dC-to-araF-C substitutions result in a dramatic enhancement in i-motif stability at acidic and neutral pH[26]. Here, we show that araF-C substitutions can also promote the formation of an unprecedented dimeric T(araF-C)$_5$ i-motif, which consists of two looping strands, associated by intermolecular C:C$^+$ base pairs. Interestingly, the loops connecting the two antiparallel segments of each strand are devoid of any nucleotide, which reflects that the length of one phosphate linker is sufficient to span the exceptionally narrow i-motif minor groove. Replacing the fourth araF-C residue with dC (as in ON4d) reduces the propensity for the sequence to fold into a dimeric i-motif (Supplementary Figure 4), indicating that the presence of an araF-C in the 3′-closing turn position is important for the integrity of the structure. To our knowledge, this study is the first to report on i-motifs with zero-nucleotide loops—a feature that highlights the unique folding requirements and capabilities of i-motifs, compared to other noncanonical structures such as G-quadruplexes. While it is in unclear whether such

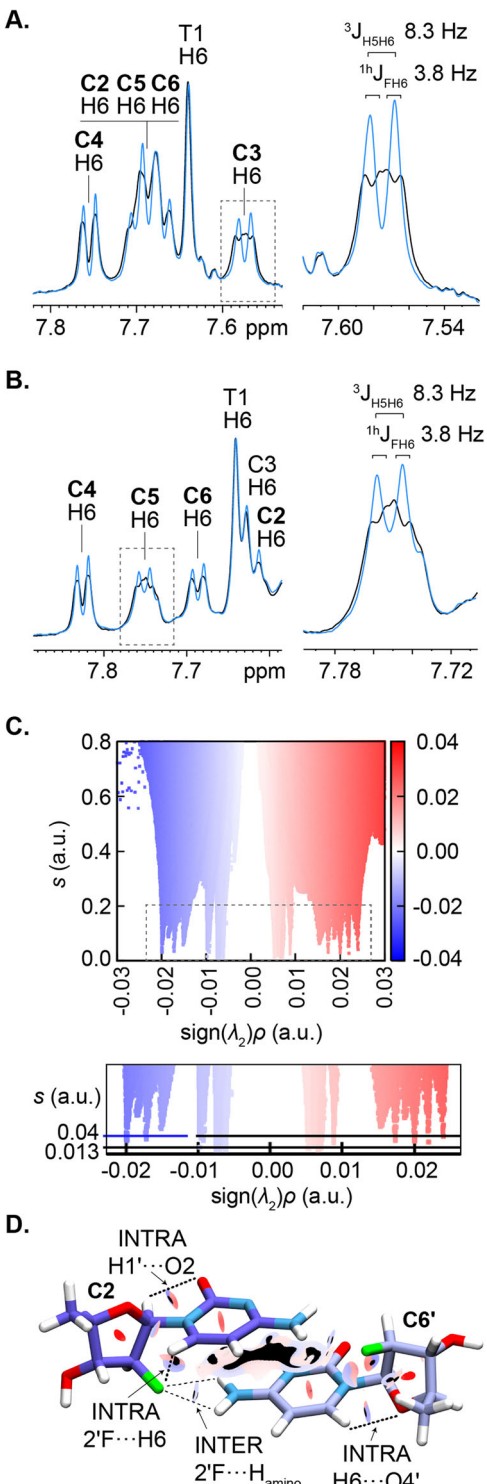

**Fig. 5 Analysis of hydrogen bonds in ON5 dimeric i-motif by NMR and NCI.** Overlapped $^{19}$F-coupled (black) and $^{19}$F-decoupled (blue) 1D $^1$H-NMR spectra showing aromatic signals of **A** RA ON5 and **B** RA ON4b. (**A**- and **B**-insets): Zoomed view of the dashed square region of **A**- and **B**-left, where the coupling constant associated with signal splitting of **C3** H6 upon $^{19}$F coupling has been measured. **C** NCI plots for **C2**–**C6′** two-residue system. Hydrogen bonds can be visualized as blue spikes in the NCI plot regions at sign($\lambda_2$)*$\rho \leq -0.01$ au. **D** Model of an inter-strand, capped araF-C pair evaluated by the NCI method, showing the $s = 0.04$ isosurface in shades of red (repulsive interactions) and shades of blue (attractive interactions), the $s = 0.013$ isosurface in black, and the most relevant intra- and inter-residual interactions detected.

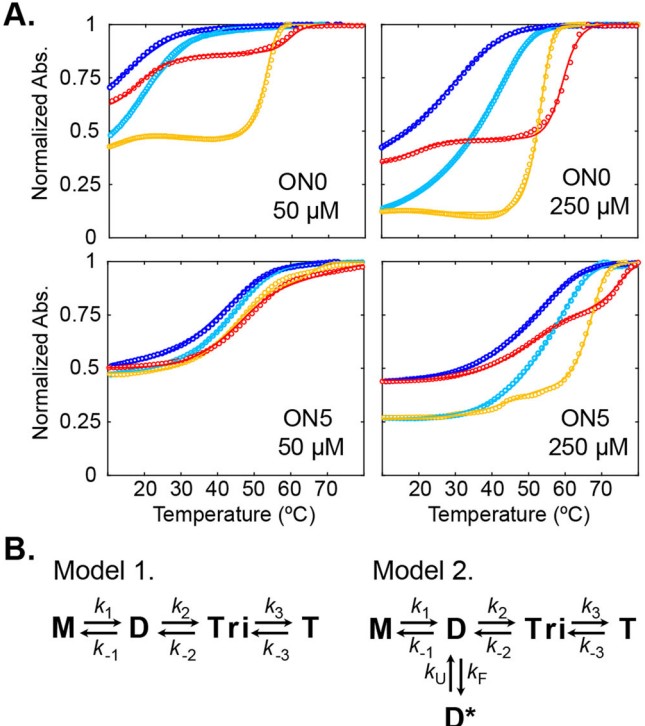

**Fig. 6 Modeling ON5 dimeric i-motif formation by thermal hysteresis analysis. A** TH traces for ON0 (top) and ON5 (bottom) at 50 μM (left) 250 μM (right). Cooling traces are shown in blue (5 °C/min) and cyan (0.5 °C/min). Heating traces are shown in red (5 °C/min) and yellow (0.5 °C/min). Experimental data are shown as points and modeled experiments corresponding to the best-fit are shown as solid lines. ON0 was fit to Model 1 and ON5 was fit to Model 2 shown in **B**. **B** Schemes of folding models. Model 1: Sequential tetrameric assembly of monomer (M) into dimer (D), trimer (Tri), and finally tetramer (T). Model 2: Sequential tetrameric assembly including an off-pathway dimeric species (D*).

loops can form in unmodified i-motifs, the $^1$H NMR spectrum of RA ON0 (Supplementary Figure 6) shows the formation of a minor species which we hypothesize is the same dimeric i-motif as ON5. Additional studies are being conducted on other cytosine-rich sequences to explore whether unmodified i-motifs could also form with zero-nucleotide loops.

In addition to determining the structure of the dimeric i-motif, we capitalized on the quality of its NMR signals (high dispersion, low overlap, and high resolution) to experimentally assess the presence of fluorine-hydrogen bonds. Previously, we attributed araF-C-induced i-motif stability to favorable electrostatic interactions based on structural information obtained for a substituted tetrameric TC$_5$ i-motif[26]. Nevertheless, the potential role of fluorine-hydrogen bonds remained an open question. The ability of organic fluorine to serve as a hydrogen bond acceptor has been debated, and only few studies have suggested the formation of fluorine-hydrogen bonds in nucleic acid systems. Often, F-H interactions have been deemed pseudohydrogen bonding, when the only experimental evidence available includes short distances (or 'contacts') and favorable angles (120-180°)[33–36]. Here, we observed $^{19}$F-induced broadening of H6 aromatic proton signals across three different i-motifs, suggesting the presence of intra-residual hydrogen bonding. While the coupling could occur through the five bonds connecting 2′F to the aromatic protons[37], we observe no splitting of signals corresponding to protons four bonds away from the 2′F, such as **C3**H4′. This corroborates our hypothesis that the coupling between 2′F and H6 is transferred through space and is a hydrogen bond in nature. In the case of

residues **C3** and **C5** of ON5 and ON4b, which adopt a north conformation, the H6 signals are split with a magnitude of 3.8 Hz (Fig. 5A, B). This *J* coupling is significantly higher than those measured between 2′F and H8 in araF-G residues in a G-quadruplex (3.0 Hz)[38] and in an araF-N:RNA hybrid duplex (2.7 Hz)[39]. Similarly, the *J* coupling reported here is more than two-fold higher than those measured for corresponding residues in araF-C (1.7 Hz) and araF-T (1.5 Hz) in the same araF-N:RNA hybrid duplex[40]. Another study in our group has shown 2′F-induced splitting of araF-G H8 protons in a fully modified araF-N duplex (araF:araF) but did not measure coupling constants[41]. Based on the remarkable splitting of the aromatic signals in **C3** and **C5** residues, we postulate that the north pucker of araF-C is conducive to stronger intra-residual fluorine-hydrogen bonds. While araF nucleosides are typically oriented in the south (C2′-endo) conformation, the pucker flexibility displayed by the araF-C residues in the dimeric i-motif reported here is not surprising[42]. In araF-N:RNA duplexes, araF residues adopt an O4′-endo (east) pucker, which provides an ideal geometry for favorable fluorine-hydrogen interactions, while minimizing steric stress[33,43]. Owing to this geometry and the resulting interactions, araF-N:RNA duplexes are superlatively stable compared to other hybrid duplexes, including DNA:RNA[43,44]. Meanwhile, araF-G residues adopting the south puckering in G-quadruplexes have favorable 2′F···H8 distances and geometries suggestive of pseudohydrogen bonds[35]. Lastly, araF-G residues have also been reported in the north pucker in a G-quadruplex with V-loop topology, thereby enabling a favorable 2′F···H-N bond[45]. To our knowledge, this study is the first to report araF pyrimidine residue oriented as north, and it is conceivable that **C3** and **C5** adopt such a geometry to optimize the stability of the dimeric i-motif.

In this study, we also use computational methods based on NCI analyses to corroborate our experimental evidence for fluorine-hydrogen bonds. To our knowledge, there are no reports that succeed in reconciling experimental and computational results to describe fluorine-hydrogen bonds in nucleic acids. A recent study showed that computations at the QTAIM-NBO (quantum theory of atoms in molecules—natural bonding orbitals) level did not provide evidence for intra-residual 2′F···H6/8 bonds in 2′-fluorinated nucleosides, despite the detection of scalar coupling by NMR[46]. However, these computational analyses are regarded as too stringent and limited to describing strong hydrogen bonds, including those with significant covalent character[28]. The NCI analyses (also based on QTAIM) used here are more suitable for describing weaker fluorine-hydrogen bonds, including those of intramolecular nature[28,47]. Consistent with the NMR data for ON5, we detect intra-residual 2′F···H6 hydrogen bonds. We also find inter-residual fluorine-hydrogen bonds between 2′F and cytosine amino protons (Fig. 4B). Unlike H6 proton signals, these amino protons do not exhibit any [19]F-induced broadening or splitting in the [1]H NMR spectrum (Supplementary Figure 20). Meanwhile, [19]F-induced splitting of inter-residual amino protons has been previously observed when araF-G was incorporated into a G-quadruplex[45]. Given that NCI analyses are based on optimized structures, it is possible that the inter-residual fluorine-hydrogen bonds detected (Fig. 5D) are very weak in nature and may not contribute significantly to the stability of the dimeric i-motif. NCI analyses also reveal non-fluorine hydrogen bonds. The O2···H-C1′ bonds detected have not been reported in nucleic acids before; it is likely that the electronegativity of 2′F makes H1′ of araF-C more electropositive, rendering it a better hydrogen bond donor. Interestingly, NCI analyses also reveal O4′···H6 hydrogen bonds; given that H6 is also hydrogen bonded intra-residually to 2′F, we hypothesize that both interactions could form a bifurcated hydrogen bond system[48,49]. Overall, we believe that the geometry adopted by

araF-C residues, optimized through their flexible puckering, is conducive to creating a network of fluorine and non-fluorine hydrogen bonds, thereby contributing to the unparalleled stability of araF-modified i-motifs.

In addition to studying the stability of araF-C modified i-motifs, we used TH analyses to study the folding mechanism of the dimeric ON5 i-motif. Several studies suggest that i-motif folding follows kinetic partitioning models[4–8,12]. Most of these studies have been carried out with monomeric i-motifs[3–5,7,8,12]. In the case of the tetrameric TC5 i-motif, the folding pathway has been previously described to proceed *via* the stepwise association of oligonucleotide strands[2]. While the dimeric and trimeric intermediates have been postulated[2], they have never been detected. Hence, aside from conformers differing in C:C[+] stacking order (3′E vs 5′E)[4,5,26], alternative i-motif folds stemming from the same sequence have never been reported. Here, we show that ON5 folding is best described by a model which leads to a kinetically-trapped off-pathway dimeric i-motif and a more thermodynamically-stable tetrameric i-motif (Fig. 6B). These strongly support the hypothesis that i-motif folding follows a kinetic partitioning mechanism. We also show that the simple sequential folding does adequately explain ON0 folding, but not ON5 folding; the latter is a significantly better fit when an off-pathway dimeric intermediate is introduced to the model. Therefore, we deduce that the araF-C modifications can change i-motif folding landscapes and can be used to sequester transient structures that would go unnoticed in the native sequence.

In conclusion, we show that the incorporation of araF-C in TC5 i-motifs slows down their folding kinetics, thus allowing the observation of metastable folding intermediates. By isolating and determining the structure of the dimeric T(araF-C)5 i-motif, we reveal the fundamental reasons for the enhanced stability provoked by araF-C substitutions in native and non-native i-motifs. While i-motifs have been used extensively in nanotechnology for their pH sensitivity[20], we demonstrate that i-motif molecularity can be modulated and controlled through rates of temperature change, which is useful for constructing novel molecular switches[50,51], thermally-tunable hydrogels[52,53], and DNA nanostructures with increased 3D complexity[54]. These i-motifs also lend to additional strand functionalization (fluorophores, cargoes…), which expands the scope of potential applications. Lastly, given that i-motifs have been recently detected in cells and that their formation seems to be transitory[14], the results of this study concretize the possibility that alternative i-motif conformations may fold in vivo and may be promoted in the absence of araF-C substitutions, through factors such as protein binding or molecular crowding effects.

## Methods

**Oligonucleotide synthesis and purification**. Oligonucleotide synthesis was performed on an ABI 3400 DNA synthesizer (Applied Biosystems) at 1 μmol scale on Unylinker (Chemgenes) CPG solid support. Thymidine (dT), deoxycytidine (N-acetyl) (dC), and deoxyadenosine (N-Bz) (dA) phosphoramidites were used at 0.1 M concentration in acetonitrile and coupled for 200 s. araF-C was used at 0.1 M concentration and coupled for 600 s. Oligonucleotide deprotection and cleavage from the solid support were achieved using aqueous AMA (30% ammonium hydroxide/40% methylamine, 1:1) at 65 °C for 1 h. Oligonucleotide sequences were purified by anion exchange HPLC on a Waters 1525 instrument using a Source 15Q Resin column (11.5 cm × 3 cm). The aqueous buffer system consisted of solution A (25% acetonitrile, 15 mM sodium acetate) and solution B (0.5 M lithium perchlorate, 25% acetonitrile, 15 mM sodium acetate) at a flow rate of 10 mL/min. The gradient was 0−100% lithium perchlorate over 50 min at 60 °C. Under these conditions, the desired peaks eluted at roughly 23 min. The purified oligonucleotides were desalted using Glen-pack desalt columns (Glen Research), and their masses were confirmed by high-resolution LC-MS.

**Annealing conditions**. Rapid or 'snap-cool' annealing (RA) involves heating the sample to 90 °C over 5 min and placing it immediately afterwards on ice for 10 min.

Slow-annealing (SA) involves heating the sample to 90 °C over 5 min, cooling it over 3 h to room temperature, and placing it in the fridge overnight.

**Native polyacrylamide gel electrophoresis (PAGE).** Oligonucleotide samples were analyzed using native gels consisting of acrylamide/bis 19:1 (20%), 10 mM sodium phosphate pH 5, and 1× TAE (Tris-Acetate-EDTA). The final gel mix solution was adjusted to pH 5 prior to casting.

Oligonucleotide samples (range of 50–200 μM, as specified in the main text) were annealed in 10 mM sodium phosphate pH 5, with a final temperature of 5 °C (by either snap-cooling or slow-annealing methods). They were mixed with 50% glycerol to attain a final glycerol concentration of ~11.5% before loading them in the gels.

After casting the gels and loading them with sample, they were run at 19 V/cm over 2 h and 7 °C, using 1× TAE pH 5 as the running buffer. Gel results were visualized by UV-shadowing. The oligonucleotide controls were dT$_{12}$ and dT$_{24}$ strands. Xylene cyanol and bromothymol blue dyes were used to monitor the progress of electrophoresis.

**Circular dichroism (CD) spectroscopy.** CD studies were performed on a Chirascan VX spectrometer using a 1 mm path-length cuvette. Temperature was maintained at 5 °C (unless otherwise specified) using the Peltier unit within the instrument. Spectra were recorded from 320–220 nm at a scan rate of 100 nm/min with three acquisitions recorded for each spectrum. The buffer spectrum was subtracted from each sample's spectrum, and the sample spectra were consequently smoothed using the Savitsky-Golay function within the Chirascan graphing software. Oligonucleotide solutions for CD measurements were 100 μM in concentration and were prepared in 10 mM sodium phosphate buffer pH 5.0. Acquisitions were conducted after snap-cooling or slow-annealing conditions.

**NMR spectroscopy.** Samples for NMR experiments were dissolved in 10 mM sodium phosphate aqueous buffer containing 10 % D$_2$O. The pH of the samples was adjusted to five by adding concentrated solutions of HCl. Oligonucleotide concentrations were 150 μM, except for SA ON4b and HC-3 samples which were at 500 μM. All NMR spectra were acquired on Bruker spectrometers operating at 600 MHz and having cryoprobes with $^1$H and $^{19}$F channels. NMR data were processed using TOPSPIN software. One-dimensional $^1$H-NMR was acquired using excitation sculpting for water suppression. Decoupling of $^{19}$F in 1D $^1$H-NMR experiments was done on the same pulse program than coupled but applying a 180° radiofrequency pulse along acquisition. NOESY spectra were acquired at mixing times of 150 and 250 ms. NOESY and DQF-COSY experiments used excitation sculpting for water suppression and a selective pulse during acquisition for $^{19}$F decoupling. TOCSY spectra were recorded with the DIPSI-2 sequence and a mixing time of 80 ms. $^1$H-$^{19}$F-HOESY experiments were set with detection in $^{19}$F, decoupling in both dimensions, and a mixing time of 150 ms. The software NMRFAM-Sparky was used for the assignment of NOESY and HOESY cross-peaks and quantitative evaluation of the signal intensities[55].

**NMR restraints for structure determination.** Distance restraints were obtained from intensities of the signals in the NOESY and HOESY spectra. NOE and HOE intensities were classified as very strong, strong, medium, weak or very weak, and distances were restrained to 3, 3.5, 4, 4.5 or 5 Å, respectively. In addition to these restraints derived from NMR spectra, target values for distance and angles related to hydrogen bonds in base pairs were set to values obtained from related structures determined by X-ray crystallography[56]. Force constants were set to 29 kcal/mol·Å$^2$ and 20 kcal/mol·Å$^2$ for base pairs hydrogen bonds and experimental distance restraints, respectively. The angular restraints for dihedral angles were obtained from qualitative analysis of intra-residual H1'-H2' (for araF-C residues) and H1'-H2' and H1'-H2'' (for dT residue) cross-peaks in the DQF-COSY spectra. Loose values of v0, v1 and v2 were set to restrain deoxyribose and 2'F-arabinose conformation to north or south domains which is equivalent to restraining the sugar pseudorotation phase angles from 0° to 36° for north conformation and between 144° and 180° for south conformation. H6-F2' distance and C6-H6···F2' angle of residues **C3** and **C5** were set to 2.0 Å and higher than 100°, respectively.

**Structure determination.** Three oligonucleotide structures were initially calculated with the program CYANA 3.0[57] and used as starting points for further refinement with the SANDER module of the molecular dynamics package AMBER 18[58]. Each refined structure was placed in the center of a cubic water box, such as to ensure a minimum distance of 8 Å from any solute atom and the edge of the box. Five sodium counterions were added to neutralize the total charge of the system. The BSC1 force field[59] and suitable parameters for araF-C were used to describe the oligonucleotide. The TIP3P model was used to simulate water molecules[60]. Hemiprotonated C:C$^+$ base pairs were modeled as base pair formation between neutral and protonated cytosines, the parameters of which are included in the BSC1 forcefield.

The system was minimized through 4000 steps of steepest descent algorithm, followed by 16,000 steps of conjugate gradient. The cartesian coordinates of the oligonucleotide were restrained applying a harmonic potential of 100 kcal/mol·A$^{-2}$. The system was slowly heated at constant volume from 0 to 298.15 K using the

Berendsen thermostat[61], and keeping the same cartesian restraints as for the minimization. Thus, five NPT equilibrations of 20 ps each were performed, during which the cartesian restraints on the oligonucleotide atoms were decreased from 100 to 5 kcal/mol·A$^{-2}$. Finally, 5 ns restrained Molecular Dynamics (rMD) were run using the NMR restraints described in the previous section.

Equilibration and rMD runs were simulated at the pressure of 1 bar and temperature of 298.15 K, applying the Berendsen barostat and thermostat algorithms[61]. For all simulations, an integration step of 2 fs was used, and the SHAKE algorithm was applied to constrain the bonds involving hydrogen atoms. Particle Mesh Ewald method was used to evaluate long-range electrostatic interactions[62]. The same protocol was applied to all three simulations starting from the different CYANA structures. Final structures were obtained by extracting ten structures from the rMD trajectories and further relaxation of the structures keeping the same restraints used during rMD simulations. Analysis of the representative structures was carried out with the programs MOLMOL[63], x3DNA[64], and Pymol[65].

**Quantum Mechanics (QM) calculations.** Five representative structures, obtained as described above, were used to evaluate the potential presence of 2'F···H intra- and inter-residual hydrogen bond. To do so at an accessible computational cost, we built two model systems for each of the five structures: one consisting of araF-cytosines **C2**, **C6**, **C2'**, **C6'** (" ' " denotes the strand containing the protonated araF-Cs) and another one comprising araF-cytosines **C3**, **C5**, **C3'**, **C5'**. The PO$_4^{2-}$ phosphate groups at the 3' and 5' positions were removed in the four nucleotides, thus the sugar 3' ends were capped with a hydroxyl group, while the 5' ends with a hydrogen atom (Supplementary Figure 14). **C2-C2'**, **C6-C6'** and **C3-C3'**, **C5-C5'** base pair hydrogen bonds, as well as base-stacking interactions, contribute to the stability of the two model systems.

**T1** and **T1'** were excluded from QM calculations as they lack fluorine and amino group and, consequently, cannot form either intra- or inter-residual 2'F···H hydrogen bonds. **C4** and **C4'** were also excluded as the position of its fluorine atom rules out the formation of any inter-residual 2'F···H hydrogen bond.

All the model systems were optimized using Gaussian 16 Rev. B.01 electronic structure software[66]. To do so, the M06-2X[67] functional and the 6–31 G(d) basis set were applied. During the optimization the positions of the O3' and C5' atoms were kept frozen. The optimized structures were used to perform a non-covalent interaction (NCI) analysis and detect intra- and inter-residual 2'F···H hydrogen bonds. The NCI approach is a method derived from the "atoms in molecules" (AIM) theory that defines chemical bonds on the basis of topological characteristics of $\rho$[68]. The quantity evaluated by the NCI analysis is $s$, where

$$s = \frac{|\nabla\rho|}{C_F * \rho^{4/3}}, \text{ where } C_{F=}2*\left(3\pi^2\right)^{1/3}$$

The quantity $s$ will assume large values as the density tails far from the nuclei, where $\rho$ is decreasing exponentially. On the other hand, it will assume small values close to the nuclei (large density and approaching-to-zero gradient). Finally, $s$ will vanish at AIM critical points (CPs), where $\nabla\rho = 0$, but also at so-called "non-AIM-CPs", in which $\nabla\rho \approx 0$[31]. If the density of two atoms overlaps, e.g., in the presence of covalent bonds, the total density will exhibit a typical AIM-CP. On the other hand, if the density overlaps in a region in which the exponential decay is asymptotic, the total density may feature a non-AIM-CP, indicating the presence of NCIs[31].

Pairs of residues containing the following cytosines were extracted from previously optimized structures: **C2-C6'**, **C3-C5'**, **C6-C2'** and **C5-C3'**. For each of them, the electronic wave function was evaluated using the M06-2X functional and the 6-311 + G(d) basis set. NCI plots, reporting the reduced density gradient ($s$) as a function of sign($\lambda_2$)$_*\rho$ (where $\lambda_2$ is the second eigenvalue of the Hessian of $\rho$), were calculated using NCIPLOT4[69]. Note that the sign($\lambda_2$) is negative for attractive interactions and positive for repulsive ones. Multiplying $\rho$ by sign($\lambda_2$) is particularly useful in the presence of hydrogen bonds (sign($\lambda_2$) < 0), which may otherwise overlap with repulsive steric clashes (sign($\lambda_2$) > 0) if plotting $s$ only as a function of $\rho$. $\rho$ values close to zero are typical of weak dispersion interactions, while hydrogen bonds are generally characterized by larger values[47].

**UV-Vis spectroscopy and global analysis of TH profiles.** UV-based thermal denaturation experiments were performed on a Cary 100 UV-Vis spectrophotometer equipped with a Peltier temperature controller. Samples were prepared in 10 mM sodium phosphate buffer (pH 5) at either 50 or 250 μM concentrations, and acquisitions were performed at 265 nm and in 1 mm path-length cuvettes. Absorbance values were acquired between 5 and 85 °C and at two scanning rates: 0.5 °C/min and 5.0 °C/min. The total change in absorbance was dependent on the scanning rate applied, we hypothesize that this is due to the dimeric species having a different absorption coefficient than the tetrameric species, and for the purpose of this analysis we assumed that the dimer had an absorption coefficient equal to half of the tetrameric species and that the trimer had an absorption coefficient equal to three-quarters of the tetrameric species. The TH profiles were globally fit to different models for the i-motif folding pathway. The mathematical relationships and assumptions made are elaborated upon in the Supplementary Information. Kinetic and thermodynamic parameters obtained from TH traces are in Supplementary Table 5.

**Reporting summary**. Further information on research design is available in the Nature Portfolio Reporting Summary linked to this article.

## Data availability
Atomic coordinates and structure factors for the reported nucleic acid structure have been deposited with the Protein Data bank under accession number 7ZYX.

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

## Acknowledgements

We acknowledge Dr. Javier López Prados, the manager of the NMR facility at the Biomolecular Interactions Platform at cicCartuja (Seville, Spain) for his contribution to obtaining high-resolution $^{19}$F- and $^{1}$H-NMR data. This investigation was supported by research grants from the Spanish "Ministerio de Ciencia e Innovación" [PID2020-116620GB-I00, to C.G., RTI2018-096704-B-I00, and PID2021-122478NB-I00 to M.O.], the Center of Excellence for HPC H2020 European Commission, "BioExcel-2 Centre of Excellence for Computational Biomolecular Research" [823830], the European Regional Development Fund under the framework of the ERFD Operative Program for Catalunya, the Catalan Government AGAUR [SGR2017-134], the European Union Marie Sklodowska Curie Action [799693, to M.G.], the Canadian Natural Sciences and Engineering Research Council of Canada (Discovery Grant, to M.J.D. and A.K.M.), and the Fonds de Recherche du Québec—Nature et Technologies Doctoral Scholarship (DE, to R.E.K). The IRB Barcelona is the recipient of a Severo Ochoa Award of Excellence from the MINECO. M.O. is an ICREA Academy scholar.

## Author contributions

R.E.K. and M.G. designed the concept of the manuscript and wrote it. R.E.K. prepared the oligonucleotide samples and characterized them by gel electrophoresis, CD spectroscopy, and UV-vis spectroscopy. V.M. carried out the computational calculations and contributed to writing and editing the manuscript. C.H. carried out thermal hysteresis analyses and kinetic simulations and contributed to writing and editing the manuscript. T.M. supervised thermal hysteresis and kinetic simulation analyses and contributed to editing the manuscript. M.O. supervised the computational analyses and contributed to editing the manuscript. C.G. supervised the project and contributed to editing the manuscript. M.G. supervised the project and carried out NMR spectroscopy and structure determination. M.J.D. supervised the project and contributed to editing the manuscript.

## Competing interests

All authors declare no competing interest.
