## [Peer Review File · Communications Chemistry]

Reviewers' comments:

Reviewer #1 (Remarks to the Author):

The manuscript by Garavis, Damha et al. reports on 2'-fluoroarabinocytidine-modified TC5 sequences that fold into unusual dimeric i-motifs assumed to represent trapped folding intermediates. Experiments and their analyses seem to have been competently carried out and given the interesting results in an important field of structural biology I recommend acceptance of the manuscript after some minor revisions according to the points raised below.

- 1) Concentrations for CD measurements are not indicated in Figure 2 and given as 100 micromolar in Figures S2 and S5. On the other hand, 150 micromolar are given in the Methods section of the main manuscript(?).
- 2) I would strongly recommend adding the concentrations for the NMR spectra in Figure 2C for a direct comparison with the gels in Figure 2A without having to resort to the Methods section.
- 3) The authors should consistently either use H2' or H2'' for araF-C hydrogens at C2' throughout the manuscript including figures and Supporting Information.
- 4) The authors write 'two strands associated head-to-tail' in the Abstract but 'two tightly turned strands, associated head-to head' on page 6 (line 124); please correct.
- 5) Sugar conformational analysis is quite relevant here, suggesting that residues in a north conformation exhibit stronger intraresidual 2'F-H-bond interactions. Therefore, the authors should elaborate more on their qualitative analysis of DQF-COSY crosspeaks (Figure S7). Are relationships between sugar pucker and approximated H1'-H2' couplings (apparently small couplings for C2 and C4 result in a cancellation of antiphase components) based on values for arabinose or preferably on 2'-fluoroarabinose nucleosides (references)?
- 6) Reference 38 is wrong and should be replaced by ChemBioChem 2018, 19, 927-930 and/or by J. Phys. Chem. Lett. 2017, 8, 5148-5152.
- 7) Page 13, line 310/311: A north conformation for an araF residue was already reported for araF-G in a quadruplex V-loop, see Chem. Commun. 2020, 56, 4539-4542.
- 8) The authors suggest the formation of interresidual F-H-N hydrogen bonds between 2'F and cytosine amino groups based on a NCI analysis but failed to see any experimental evidence when looking at corresponding amino proton resonances (Figure S20). It should be mentioned that such hydrogen bonds have been reported before in G-quadruplexes and experimentally verified by a noticeable decrease in linewidth of 4-5 Hz upon 19F decoupling for amino protons with non-resolved splittings (Chem. Commun. 2020, 56, 4539-4542). The lack of changes in linewidths with 19F decoupling in the present study (Figure S20) does not corroborate such hydrogen bonds to be relevant. The authors should comment on this.

Reviewer #2 (Remarks to the Author):

Thank you for the opportunity to read this well-written and conducted study. The manuscript is titled "2'-Fluoroarabinocytidine traps i-motif folding intermediates through 2 fluorine-hydrogen bonds". The work describes i-motif folding and trapping of intermediates through using a range of spectroscopic methods. Though, almost hidden within (certainly not obvious from the title) is actually a gem of information that is entirely unique - the first observation of i-motifs with zero nucleotide loops. This is an exceptional observation and would be of the utmost interest to people studying nucleic acids. This further shows how different i-motifs are compared to G-quadruplexes and alters the potential folding capabilities of these structures. One does ask the question about whether these are possible in non-modified nucleic acids. Do the authors have any evidence of this? Or is this

an artefact, unique to these particular modified oligos? If this is possible in the unmodified analogues then this opens up the folding algorithms for i-motif in a completely different way to G-quadruplex. It also would explain why it is difficult to stop i-motif formation - i.e. it is easy to make them weaker/less stable, but very difficult to stop any i-motif from forming.

Overall the manuscript is exceptionally well written and the methods used are cutting edge and appropriate. I recommend publishing but would suggest that the authors include the observation of 0 nt loops in the title or incorporate it in there somewhere - this will increase the impact of this paper significantly:

*Suggest changing the title to include the fact that 0 nt loops have been observed - this is the detail that people will cite and use this paper for - above all other components

*Include the title of the paper in the SI

*Include **all** details in the figure captions e.g. buffer, concentrations, pH etc. e.g. Fig 2

*The normalised absorbances on the UV spectra do not make sense as they result in negative absorbances. Normalising should be the equivalent of conversion to a fraction folded (i.e. 0 to 1 from folded to unfolded). The normalisation here doesn't make sense when it is not possible to have a negative absorbance. Do these even need to be normalised?

*Discussion about the zero nucleotide loops and whether this feature is relevant to non-modified oligos would be helpful for the reader - as would any additional evidence that this is possible in non-modified oligos. Currently the discussion only includes reference to forming the dimer - and is not specific about the 0 nt loops in particular.

Reviewer #3 (Remarks to the Author):

The manuscript entitled "2'-Fluoroarabincytidine traps i-motif folding intermediates through fluorine-hydrogen bonds" describes the structural determination and kinetic folding profile of the T-(araF-C)₅ i-motif. The subject is clearly relevant to the audience of the journal and the reported results are valuable.

However, the manuscript should be accepted only after addressing the following points.

1) It is stated that the stabilization of the i-motif is due mainly to the formation of inter- and intra-residual hydrogen bonds with fluorine. It should be pointed out that the presence of the fluorine has also an important role in polarizing adjacent bonds. Specifically, the mentioned O2...H1' hydrogen bond could also play an important role in the stabilization due to the increased acidity character of the H1' originating from the strong electron withdrawing effect of the fluorine in alpha position. The authors should comment this relevant point.

2) In the legend of Figure 2, the authors should provide the concentration used for acquiring the 1H NMR spectra.

3) In Figure 3b, the authors should comment why in the 19F NMR spectrum of ON4b only three fluorine signals are visible, as one would expect to see four signals.

4) In Figure 3c, the authors should comment why the cross peaks between the 2'F of C6 and the C2H4-1 and C2H4-2 are not visible.

5) In Figure 5d, it seems that, in addition to the close distance and NCI between the H6 and fluorine atom, there is also the presence of a NCI between the H6 and the oxygen of the ribose. If this is the case, please add a comment as this interaction could also be relevant.

6) Across the manuscript the authors should be consistent in calling the interactions involving the fluorine atom either pseudo-hydrogen bonds or hydrogen bonds.

Response to Reviewers:

Responses to all three reviewers are mentioned below:

Reviewer #1 (Remarks to the Author):

The manuscript by Garavis, Damha et al. reports on 2'-fluoroarabinocytidine-modified TC5 sequences that fold into unusual dimeric i-motifs assumed to represent trapped folding intermediates. Experiments and their analyses seem to have been competently carried out and given the interesting results in an important field of structural biology I recommend acceptance of the manuscript after some minor revisions according to the points raised below.

Thank you for reviewing the manuscript and for your positive feedback. We have made the minor revisions as suggested.

- 1) Concentrations for CD measurements are not indicated in Figure 2 and given as 100 micromolar in Figures S2 and S5. On the other hand, 150 micromolar are given in the Methods section of the main manuscript(?).

Thank you for pointing out this inconsistency. The values have been rectified and reflected in corresponding areas of the manuscript. The values have also been added to the revised **Figure 2**.

- 2) I would strongly recommend adding the concentrations for the NMR spectra in Figure 2C for a direct comparison with the gels in Figure 2A without having to resort to the Methods section.

The concentration has been added to the **Figure 2** caption and revised **Figure 2**.

- 3) The authors should consistently either use H2' or H2'' for araF-C hydrogens at C2' throughout the manuscript including figures and Supporting Information.

Indeed, there was an inconsistency in the names of the H atom attached to the C2' position of the araF-C residues. We thank the reviewer for the observation. We have used now the name H2' for those atoms. The changes have been highlighted in the text. We have also modified the **Table S1** and added a sentence on its description to specify the nomenclature used for such atoms.

- 4) The authors write 'two strands associated head-to-tail' in the Abstract but 'two tightly turned strands, associated head-to head' on page 6 (line 124); please correct.

Thank you for pointing out this mistake. The abstract has been modified accordingly.

- 5) Sugar conformational analysis is quite relevant here, suggesting that residues in a north conformation exhibit stronger intraresidual 2'-F-H-bond interactions. Therefore, the authors should elaborate more on their qualitative analysis of DQF-COSY crosspeaks (Figure S7). Are relationships between sugar puckers and approximated H1'-H2' couplings (apparently small couplings for C2 and C4 result in a cancellation of antiphase components) based on values for arabinose or preferably on 2'-fluoroarabinose nucleosides (references)?

To determine the sugar pucker, we relied on the DQF-COSY spectrum. To analyze the spectrum, we used the Karplus curve relating the pseudorotational angle (P) with $J_{H1'-H2'}$ obtained from solving the Karplus equation (S. S. Wijmenga, M. M. W. Mooren and C. W. Hilbers, in *NMR of Macromolecules. A Practical Approach*, ed. G. C. K. Roberts, IRL Press, New York, 1993.) for a 2'-F-arabinose. The curve shows that the magnitude of $J_{H1'-H2'}$ is remarkably smaller in a range of P corresponding to the South conformation (P between 144° and 180°) which results in the cancellation of the H1'-H2'

COSY signals of the residues with South puckering. We have added a new panel in former **Figure S7 (now Figure S8)** showing the Karplus curve used for the analysis. For comparison, we have also included, in the same plot, the Karplus curve of the equivalent J ($J_{H1'-H2'}$) for deoxyribose. The description of the analysis of the DQF-COSY to determine the sugar puckering of araF-C residues has been added in the caption of **Figure S8**.

- 6) Reference 38 is wrong and should be replaced by ChemBioChem 2018, 19, 927-930 and/or by J. Phys. Chem. Lett. 2017, 8, 5148-5152.

Thank you for pointing this out. Reference 38 was indeed wrong and has been replaced by both references suggested by the reviewer.

- 7) Page 13, line 310/311: A north conformation for an araF residue was already reported for araF-G in a quadruplex V-loop, see Chem. Commun. 2020, 56, 4539-4542.

Thank you for pointing this out to us. Indeed, araF-G residue has been reported in the North conformation in the G-quadruplex with conventional V-loop and we have added the following sentence in the second paragraph of the Discussion section:

“Lastly, araF-G residues have also been reported in the North pucker in a G-quadruplex with V-loop topology, thereby enabling a favorable 2'F...H-N bond.”⁴⁶ (Chem. Commun. 2020, article cited).

- 8) The authors suggest the formation of interresidual F-H-N hydrogen bonds between 2'F and cytosine amino groups based on a NCI analysis but failed to see any experimental evidence when looking at corresponding amino proton resonances (Figure S20). It should be mentioned that such hydrogen bonds have been reported before in G-quadruplexes and experimentally verified by a noticeable decrease in linewidth of 4–5 Hz upon ¹⁹F decoupling for amino protons with non-resolved splittings (Chem. Commun. 2020, 56, 4539-4542). The lack of changes in linewidths with ¹⁹F decoupling in the present study (Figure S20) does not corroborate such hydrogen bonds to be relevant. The authors should comment on this.

We thank the reviewer for pointing our attention to the experimentally verified decreased in linewidth of 4-5 Hz upon ¹⁹F decoupling for amino protons with non-resolved splittings. Indeed, while we had detected inter-residual hydrogen bonds between ¹⁹F and amino protons *via* NCI, we did not detect them experimentally *via* NMR. Since NCI methods use optimized structures, it is reasonable to think that while such interactions are detected for certain structures, they may be less relevant when considering the solution structures studied in bulk by NMR.

As a result, we have removed the following postulate on why we do not detect the splitting by NMR: “which may be explained by the broad linewidths and chemical exchange rates of amino protons with solvent” and we have also amended the text as follows:

“Meanwhile, ¹⁹F-induced splitting of inter-residual amino protons has been previously observed when araF-G was incorporated into a G-quadruplex.⁴⁶ Given that NCI analyses are based on optimized structures, it is possible that the inter-residual fluorine-hydrogen bonds detected are very weak in nature and may not contribute significantly to the stability of the dimeric i-motif.”

Reviewer #2 (Remarks to the Author):

Thank you for the opportunity to read this well-written and conducted study. The manuscript is titled "2'-Fluoroarabincytidine traps i-motif folding intermediates through fluorine-hydrogen bonds". The work describes i-motif folding and trapping of intermediates through using a range of spectroscopic methods. Though, almost hidden within (certainly not obvious from the title) is actually a gem of information that is entirely unique - the first observation of i-motifs with zero nucleotide loops. This is an exceptional observation and would be of the utmost interest to people studying nucleic acids. This further shows how different i-motifs are compared to G-quadruplexes and alters the potential folding capabilities of these structures. **One does ask the question about whether these are possible in non-modified nucleic acids. Do the authors have any evidence of this? Or is this an artefact, unique to these particular modified oligos? If this is possible in the unmodified analogues then this opens up the folding algorithms for i-motif in a completely different way to G-quadruplex. It also would explain why it is difficult to stop i-motif formation - i.e. it is easy to make them weaker/less stable, but very difficult to stop any i-motif from forming.**

We would like to thank the reviewer for the positive feedback and for asking such interesting questions pertaining to our work. We agree that an elaboration on the zero-nucleotide loop characteristic of the dimeric i-motif would be pertinent to the discussion in this manuscript. We do not have direct evidence to establish that the i-motif with zero-nucleotide loops can form in the unmodified sequence. However, the ¹H NMR spectrum of rapid-annealed ON0 (**Figure S6**) shows the formation of a minor species which we hypothesize could be the same dimeric i-motif with zero-nucleotide loops. We are currently investigating additional sequences to explore the formation of similar i-motifs in the absence of any cytidine modifications.

Overall the manuscript is exceptionally well written and the methods used are cutting edge and appropriate. I recommend publishing but **would suggest that the authors include the observation of 0 nt loops in the title or incorporate it in there somewhere** - this will increase the impact of this paper significantly:

- 1) Suggest changing the title to include the fact that 0 nt loops have been observed - this is the detail that people will cite and use this paper for - above all other components

We have considered this suggestion and modified the title accordingly: "i-Motif folding intermediates with zero-nucleotide loops are trapped by 2'-fluoroarabincytidine *via* F...H and O...H hydrogen bonds"

- 2) Include the title of the paper in the SI

We have included the title in the SI.

- 3) Include *all* details in the figure captions e.g. buffer, concentrations, pH etc. e.g. Fig2.

We have modified **Figure 2** caption to include all details necessary for interpretation.

- 4) The normalised absorbances on the UV spectra do not make sense as they result in negative absorbances. Normalising should be the equivalent of conversion to a fraction folded (i.e. 0 to 1 from folded to unfolded). The normalisation here doesn't make sense when it is not possible to have a negative absorbance. Do these even need to be normalised?

We thank the reviewer for their comment. The y-axis of **Figure 6A** actually currently represents 'Normalized Δ Absorbance' with values between -1 and 0. To avoid confusion, we have changed the y-axis to 'Normalized Absorbance' with values between 0 and 1.

- 5) Discussion about the zero nucleotide loops and whether this feature is relevant to non-modified oligos would be helpful for the reader - as would any additional evidence that this is possible in non-modified oligos. Currently the discussion only includes reference to forming the dimer - and is not specific about the 0 nt loops in particular.

Following the reviewer's suggestion, we have added the following to the discussion:

"To our knowledge, this study is the first to report on i-motifs with zero-nucleotide loops – a feature that highlights the unique folding requirements and capabilities of i-motifs, compared to other noncanonical structures such as G-quadruplexes. While it is unclear whether such loops can form in unmodified i-motifs, the ^1H NMR spectrum of RA ON0 shows the formation of a minor species which we hypothesize is the same dimeric i-motif as ON5. Additional studies are being conducted on other cytosine-rich sequences to explore whether unmodified i-motifs could also form with zero-nucleotide loops."

Reviewer #3 (Remarks to the Author):

The manuscript entitled “2'-Fluoroarabincytidine traps i-motif folding intermediates through fluorine-hydrogen bonds” describes the structural determination and kinetic folding profile of the T-(araF-C)₅ i-motif. The subject is clearly relevant to the audience of the journal and the reported results are valuable.

However, the manuscript should be accepted only after addressing the following points.

- 1) It is stated that the stabilization of the i-motif is due mainly to the formation of inter- and intra-residual hydrogen bonds with fluorine. It should be pointed out that the presence of the fluorine has also an important role in polarizing adjacent bonds. Specifically, the mentioned O2...H1' hydrogen bond could also play an important role in the stabilization due to the increased acidity character of the H1' originating from the strong electron withdrawing effect of the fluorine in alpha position. The authors should comment this relevant point.

Thank you for your comment. We agree that this point is relevant and we modified the corresponding sentence in the discussion as follows: “The O2···H-C1' bonds detected have not been reported in nucleic acids before; it is likely that the electronegativity of 2'F makes H1' of araF-C more electropositive, rendering it a better hydrogen-bond acceptor.”

- 2) In the legend of Figure 2, the authors should provide the concentration used for acquiring the 1H NMR spectra.

The captions of **Figure 2** have been adjusted accordingly. The concentrations have also been provided in Revised **Figure 2** panels.

- 3) In Figure 3b, the authors should comment why in the 19F NMR spectrum of ON4b only three fluorine signals are visible, as one would expect to see four signals.

Thank you for pointing this out. Actually, the HOESY spectrum shows that there are indeed four signals. Two of them appear very close together. We have included the HOESY spectrum in the supporting information and added the following sentence to the text:

“In ON4b specifically, two ¹⁹F signals appear close together at -193.10 and 193.14 ppm and are resolved in the ¹H-¹⁹F-HOESY (**Figure S7**).”

- 4) In Figure 3c, the authors should comment why the cross peaks between the 2'F of C6 and the C2H4-1 and C2H4-2 are not visible.

Most probably, these cross-peaks are not visible because of two factors. First, **C2:C2⁺** is the terminal hemiprotonated base pair and thus it is more exposed to the solvent. Therefore, the exchange rates of its exchangeable protons (imino and amino) are presumably higher. This provokes that its imino proton and the cross-peaks involving the amino protons (as the HOE with 2'F of **C6**) are not visible.

In addition, **C6** is the 3'-terminal residue and probably more dynamic than the others. In fact, the average **C2**amino – **C6**2'F distance is slightly larger than other inter-residual amino – F distances.

We have made changes in the text as follows:

“The same correlations are observed between **C2** 2'F and **C6** amino protons, but they are not observed between **C6** 2'F and **C2** amino protons, likely because the **C2:C2⁺** base

pair is the most external, with its amino protons exchanging more rapidly with solvent. We also detect intense HOE cross-peaks correlating 2'F and H6 aromatic protons of all araF-C residues (**Figure 3C**)."

- 5) In Figure 5d, it seems that, in addition to the close distance and NCI between the H6 and fluorine atom, there is also the presence of a NCI between the H6 and the oxygen of the ribose. If this is the case, please add a comment as this interaction could also be relevant.

We thank the reviewer for bringing our attention to this interaction, which indeed has been missed in our analysis. We comment on the relevance of this interaction in the discussion as follows:

"Interestingly, NCI analyses also reveal O4'...H6 hydrogen bonds; given that H6 is also hydrogen bonded intra-residually to 2'F, we hypothesize that both interactions could form a bifurcated hydrogen bond system.^{49,50}"

We also highlight this interaction in revised **Figure 5D** and make minor textual changes to reflect the relevance of these interactions.

- 6) Across the manuscript the authors should be consistent in calling the interactions involving the fluorine atom either pseudo-hydrogen bonds or hydrogen bonds.

Thank you for bringing this up. We use the term 'Pseudo-hydrogen bonds' only when referring to cited works which use this term. We also explain in the manuscript that 'pseudo-hydrogen bonds' is often a term given "when the only experimental evidence available includes short distances (or 'contacts') and favorable angles (120-180°).³³⁻³⁶"

REVIEWERS' COMMENTS:

Reviewer #1 (Remarks to the Author):

I'm fully satisfied with the revisions and have no further objections.

Reviewer #2 (Remarks to the Author):

Thank you to the authors who have made significant revisions to the manuscript. I recommend that it is accepted.

Reviewer #3 (Remarks to the Author):

Dear Authors, Regarding the answer to Reviewer #3 - point 1) below:

I think you made a typo. I suppose you wanted to say that the fluorine atom in alpha to the H1' makes the hydrogen a better hydrogen bond donor and not a better hydrogen bond acceptor.

Response to Reviewers:

Reviewer #1 (Remarks to the Author):

I'm fully satisfied with the revisions and have no further objections.

Thank you.

Reviewer #2 (Remarks to the Author):

Thank you to the authors who have made significant revisions to the manuscript. I recommend that it is accepted.

Thank you.

Reviewer #3 (Remarks to the Author):

Dear Authors, Regarding the answer to Reviewer #3 - point 1) below:

I think you made a typo. I suppose you wanted to say that the fluorine atom in alpha to the H1' makes the hydrogen a better hydrogen bond donor and not a better hydrogen bond acceptor.

Thank you, we have addressed the typo.